# The self-healing of defects induced by the hydriding phase transformation in palladium nanoparticles

A. Ulvestad [1] & A. Yau [2]

Nanosizing can dramatically alter material properties by enhancing surface thermodynamic contributions, shortening diffusion lengths, and increasing the number of catalytically active sites per unit volume. These mechanisms have been used to explain the improved properties of catalysts, battery materials, plasmonic materials, etc. Here we show that Pd nanoparticles also have the ability to self-heal defects in their crystal structures. Using Bragg coherent diffractive imaging, we image dislocations nucleated deep in a Pd nanoparticle during the forward hydriding phase transformation that heal during the reverse transformation, despite the region surrounding the dislocations remaining in the hydrogen-poor phase. We show that defective Pd nanoparticles exhibit sloped isotherms, indicating that defects act as additional barriers to the phase transformation. Our results resolve the formation and healing of structural defects during phase transformations at the single nanoparticle level and offer an additional perspective as to how and why nanoparticles differ from their bulk counterparts.

[1] Materials Science Division, Argonne National Laboratory, Argonne, IL 60439, USA. [2] Department of Chemistry, Stanford University, Stanford, CA 94305, USA. Correspondence and requests for materials should be addressed to A.U. (email: aulvestad@anl.gov)

An area of active research continues to be the design and understanding of nanomaterials as they differ from their bulk counterparts in many of their properties, including mechanical, electronic, optical, and thermodynamic[1–6]. In particular, recent research has focused on the structural transformation of crystalline lattices to accommodate solute atoms, which is relevant to technologically important systems such as hydrogen storage in metals[7] and lithium storage in rechargeable batteries[8]. In these systems, the crystalline particles often undergo phase transformations between solute-rich and solute-poor phases that have a lattice mismatch, which can result in the nucleation of defects like dislocations. At the same time, nanostructuring is known to improve rate capabilities, lifetime, and reduce overpotential[3, 9]. One explanation for the improvement includes the prevention of defect nucleation via suppression of two-phase coexistence[10–12].

The hydrogen–palladium system has especially attracted interest as a model platform for studying such solute-induced strain effects. In the hydrogen–palladium system[13, 14], the maximum hydrogen solubility is reached in the hydrogen-poor phase near the upper spinodal pressure. Further increase in the hydrogen concentration is accompanied by transformations of crystal regions to the lattice-expanded hydrogen-rich phase[14, 15]. Many previous experimental studies have used electron microscopy[12, 16, 17] and plasmonic techniques[18] to track individual Pd nanoparticles during the hydriding phase transformation. Ensemble studies have been performed with luminescence-based methods[19].

Recent research has shown that the crystal quality of Pd nanoparticles can improve as a result of the phase transformation[17]. However, these studies have focused on Pd nanoparticles smaller than 80 nm, which is below the critical size required for dislocation nucleation[20]. Dislocation nucleation is driven by strain energy relaxation. The lattice mismatch between the hydrogen-rich and the hydrogen-poor phases is approximately 3.5%. As such, it becomes energetically favorable above a critical size to relieve the strain energy through the introduction of dislocations. For the idealized spherical particle with a spherical shell

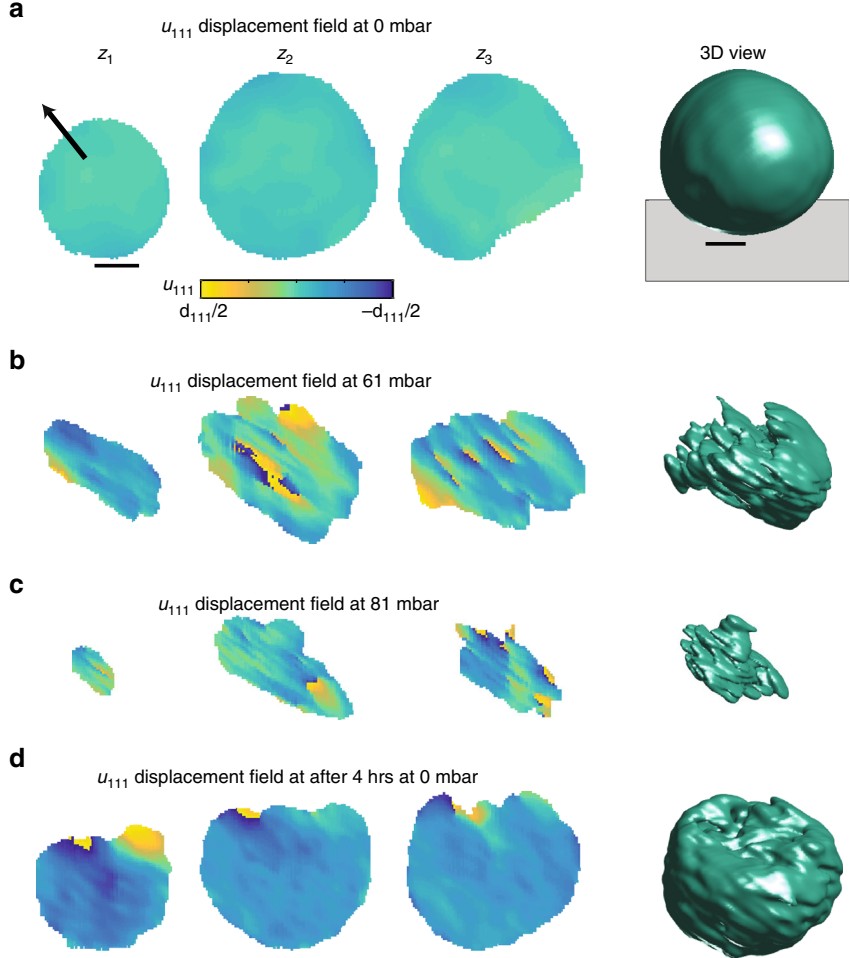

**Fig. 1** 3D Imaging of the displacement field of an individual Pd nanoparticle self-healing defects. Three cross-sections of the $u_{111}$ displacement field through the particle volume (see Supplementary Fig. 2 for their spatial location) and the 3D view are shown. The scale bars represent 100 nm. **a** The Pd nanoparticle shape in the as-synthesized state. The displacement field cross-sections $z_1$–$z_3$ show the particle is nearly strain-free and has no defects visible at this scattering condition. The black arrow shows the [111] direction. The 3D view shows the particle is approximately spherical. The grey plane in the 3D view indicates the substrate. **b** The phase transformation is driven by incrementing the hydrogen partial pressure in a step-wise fashion. At 61 mbar $pH_2$, regions that have transformed to the hydrogen-rich phase appear missing. Dislocations are induced in the hydrogen-poor phase due to the lattice mismatch between the two phases. The location of displacement field vortices corresponds to the dislocation cores. **c** Increasing $pH_2$ to 81 mbar results in more regions transforming to the hydrogen-rich phase and dislocation rearrangement. **d** The phase transformation is reversed by applying 0 mbar of hydrogen partial pressure. After 4 h, the particle has almost completely returned to the hydrogen-poor phase. More importantly, the dislocations that were induced in the particle's center have healed. Only a few dislocations exist near the top of the particle, close to the phase boundary

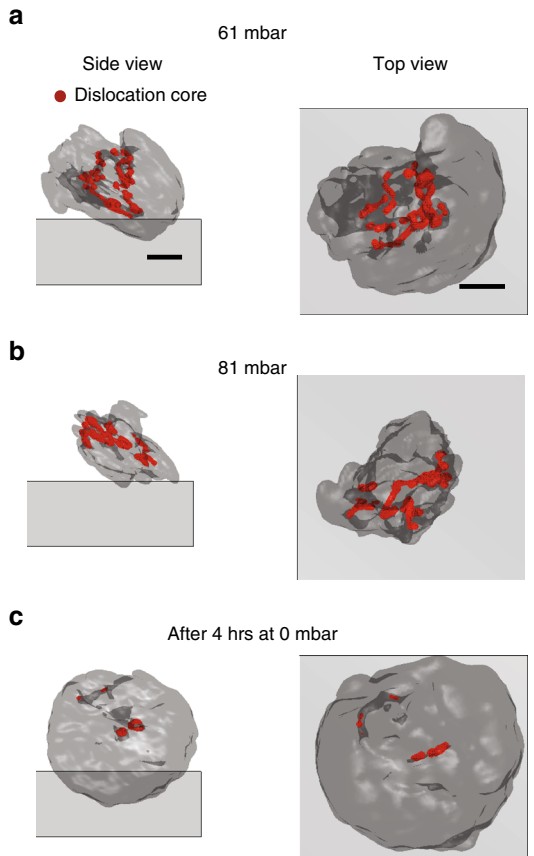

**Fig. 2** The 3D dislocation network evolution due to changes in hydrogen partial pressure. The particle shape is shown as a black semi-transparent isosurface, the substrate is shown as a grey plane, and the dislocation line is shown in red. Only the states with a dislocation are shown. The scale bars represent 100 nm. **a** The dislocation network at 61 mbar. **b** The dislocation network at 81 mbar. **c** The dislocation network after 4 h at 0 mbar

of hydrogen-rich phase, misfit dislocation loops are introduced at the phase boundary[12, 15]. However, this description is a rough approximation of the experimental observations. The two-phase morphology is more similar to a spherical cap with a planar interface between the two phases[17, 20–22]. Although the phase morphology is different, strain energy relief still drives dislocation nucleation.

Here we use the hydrogen–palladium system to showcase a previously unobserved phenomenon – the ability of Pd nanoparticles to self-heal crystallographic defects, making them more durable, strain-tolerant versions of their bulk counterparts. To do this, we utilize a relatively new X-ray imaging technique known as Bragg coherent diffractive imaging (BCDI) to track the nucleation and motion of dislocations during the partial hydriding and complete dehydriding phase transformation. We also show that the defective Pd nanoparticles exhibit sloped isotherms, suggesting that these defects act as additional barriers to the phase transformation.

## Results

**Experimental Description.** For this study, Pd nanocrystals with linear dimensions between 200 and 400 nm were formed via thermal dewetting of 25 nm thick Pd films (Supplementary Fig. 1). BCDI experiments were performed at beamline 34-ID-C of the Advanced Photon Source using 9 keV X-rays. In BCDI, the nanoparticle shape and strain are resolved in 3D detail by recording coherent diffraction patterns of a Bragg peak from a

single nanoparticle[23]. The phase of the diffraction patterns is reconstructed iteratively using phase retrieval algorithms[24]. BCDI is incredibly sensitive to crystallographic dislocations because of the large displacement fields they induce in the crystal lattice[25, 26]. After initial characterization of the selected Pd nanoparticle, variable hydrogen pressure experiments were conducted. Please see the Methods for further details. The primary difference between this work and that of ref. [20] is the amount of driving force supplied by the hydrogen partial pressure. In ref. [20], the hydrogen partial pressure was very close to the upper spinodal pressure and consequently the driving force for the phase transformation was small. In this work, the partial pressure is incremented well beyond the upper spinodal pressure and consequently the driving force is large.

**Dislocation healing during the reverse phase transformation.** The as-synthesized particle is spherical with an approximate diameter of 400 nm (Fig. 1a) and reconstructed with 20–30 nm resolution (see Methods section). This size is predicted to be above the critical size necessary for dislocation nucleation[12, 20]. The displacement field (Fig. 1a, $z_1$–$z_3$) is near zero and smoothly varying, consistent with a nearly strain-free particle. See Supplementary Fig. 2 for the spatial locations of the cross-sections. In slice $z_3$, the flat facet at the bottom right could indicate a coherent twin boundary[27] as the particles appear very spherical in the SEM images. The black vector shows the [111] direction and the facet it exits is thus a {111} facet. Most importantly, the displacement field is free of any spiral regions in which the displacement field varies from $-d_{111}/2$ to $+d_{111}/2$ (displacement field vortices) and thus the particle has no dislocations that are visible at this scattering condition. For a dislocation to be visible at a particular scattering vector, there must be a non-zero projection of the displacement field onto the scattering vector. Additionally, defects such as vacancy and interstitial clusters are below the measurement resolution.

The phase transformation is driven in a step-wise manner by incrementing the externally applied hydrogen partial pressure ($pH_2$). At a $pH_2$ of 61 mbar, the particle image exhibits two distinct changes (Fig. 1b) relative to the as-synthesized state. The first is the disappearance of particular regions. These regions correspond to the hydrogen-rich phase as only the signal diffracting into the hydrogen-poor phase is used for the reconstruction[22]. The second is the appearance of displacement field vortices, in particular, in cross-sections $z_2$ and $z_3$. These are due to dislocations, and it is clear that these dislocations exist in the particle center that remains in the hydrogen-poor phase. In this work, we cannot comment on the dislocation nucleation mechanism[28–30] as it occurs too rapidly relative to the experimental measurement time. However, we can observe the stability and mobility of dislocations after they are nucleated. As will be shown later using the 3D view, the dislocation lines form an extended network throughout the crystal that connects to the phase boundary.

The phase transformation is driven further by applying a $pH_2$ of 81 mbar, which results in dislocation rearrangement and further volume reduction (Fig. 1c). The phase transformation is reversed by applying 0 mbar of hydrogen partial pressure for 4 h. The particle has then almost completely returned to the hydrogen-poor phase (Fig. 1d). The dislocations that were induced in the particle's center have been annihilated. Only a few dislocations exist near the top of the particle, close to the phase boundary, the location of which can be inferred based on the missing parts of the image relative to Fig. 1a. Unfortunately, the annihilation mechanism is unclear as measurements were not made continuously during the 4 h at 0 mbar. However, future

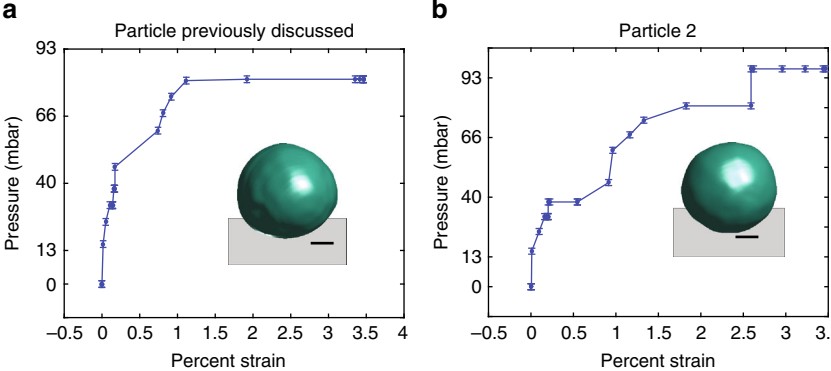

**Fig. 3** Example isotherms for particles that undergo defect nucleation. The scale bars represent 100 nm. **a** The isotherm for the particle previously discussed. Note that the point corresponding to the return to 0 mbar after 81 mbar is omitted for clarity. The isotherm exhibits sloping and more than a single plateau. **b** The computed isotherm for another particle. Again, the isotherm exhibits sloping and more than a single plateau. Error bars show the average drift in the pressure gauge over the course of the measurements

experiments will be able to address this issue using the tools and analysis developed in this work.

Finally, we drove the phase transformation to completion by again applying a $pH_2$ of 81 mbar. Reconstructions of the coherent diffraction data after the complete transformation (Supplementary Fig. 3) are not reproducible due to the complexity of the data, which implies that the particle has a high density of defects such as dislocations. If the particle was able to heal defects during the complete transformation, we would expect a coherent diffraction pattern very similar to the pattern of the initial state but this is not the case.

The appearance and subsequent disappearance of dislocations in the particle center, which remains in the hydrogen-poor phase, is remarkable. Previously, the damage induced during the phase transformation was expected only to be healed by heating the lattice to high temperatures[31]. Fig. 1 shows this is not the case for nanoparticles. Defect generation results in significant changes to the coherent diffraction data that then mostly reverse during the dehydriding transformation (Supplementary Fig. 4). For another particle that exhibits the same healing mechanism, see Supplementary Fig. 5. While this particle does not relax as fully within the 4 h at 0 mbar as the particle discussed in Fig. 1, this is probably due to slight differences in morphology. Consequently, there is likely variation in the degree to which particles can remove defects.

While the 2D cross-sections clearly show the appearance and disappearance of dislocations in the particle center, BCDI can image the dislocation line in 3D detail. We construct the dislocation line using a gradient-based method to identify the essential displacement field vortices throughout the 3D image.

**Evolution of the 3D dislocation network**. As discussed previously, the lattice mismatch between the hydrogen-poor and hydrogen-rich phases induces dislocations for particles above a critical size (~ 300 nm). The dislocation network at 61 mbar is complex and extends from the top phase boundary through the particle center to the bottom of the phase boundary (Fig. 2a). The dislocation lines form an extended network throughout the crystal. Further incrementing the hydrogen partial pressure to 81 mbar results in dislocation rearrangement and further hydrogen-poor phase volume reduction (Fig. 2b). Amazingly, nearly all of the defects in the hydrogen-poor phase are healed after 4 h at 0 mbar (Fig. 2c). The remaining defects appear to have the structure of dislocation loops[32, 33] (most clear from Fig. 2c Side view). The ability to resolve the entire defect network in 3D detail shows that the dislocation lines induced during the partial forward

transformation extend throughout the particle and are subsequently self-healed during the reverse transformation. On the basis of this data, we speculate that the dislocation loops in the particle interior are able to collapse during the dehydriding phase transformation. The remaining loops near the surface could be stabilized by a dislocation loop-surface interaction. Future experiments can address the mechanism using the results of this work.

**Defects act as barriers to phase transformations**. Finally, an open question in the palladium–hydrogen system that we address is the nature of the thermodynamic isotherm after defect nucleation. To construct these isotherms from the diffraction data, we split the calculation into two pieces: one for the solid solution regime and one for the two-phase regime. In the solid solution regime before the phase transformation, the hydrogen-poor phase of the Pd lattice expands linearly with the hydrogen concentration and thus the percent strain is linearly related to the hydrogen concentration. Using the Bragg peak location to define the average lattice constant, the percent strain can then be calculated via

$$\epsilon = 100 * \frac{a_{pH_2} - a_0}{a_0} \qquad (1)$$

where $a_0$ is the initial average lattice constant before any hydrogen exposure and $a_{pH_2}$ is the average lattice constant at a given pressure. In the case of Fig. 3a, the solid solution regime lasts through 46.5 mbar $pH_2$.

Once the hydrogen-rich phase begins to appear, the average lattice constant of the hydrogen-poor phase is no longer proportional to the hydrogen concentration. In this regime, the percent strain is calculated via

$$\epsilon = 100 * \left( x_\alpha \frac{3.896 - 3.89}{3.89} + (1 - x_\alpha) \frac{4.025 - 3.89}{3.89} \right) \qquad (2)$$

where $x_\alpha$ is the fraction remaining in the hydrogen-poor phase. This is determined using the square root of the ratio of the total scattering intensity at a given partial pressure to the total scattering intensity in the initial hydrogen-free state[34]. The isotherms constructed for two different particles in this way are shown in Fig. 3.

After reversing the phase transformation by applying 0 mbar for 4 h, the hydrogen partial pressure is incremented to 81 mbar. The isotherm including the return to 0 mbar data point is shown in Supplementary Fig. 6. In nanoparticles undergoing coherent

phase transformations, the two phases do not coexist at thermodynamic equilibrium and the isotherm exhibits a single plateau near the upper spinodal pressure[12, 15]. The particles studied here are larger than the critical size for defect nucleation and exhibit sloped isotherms with multiple plateaus (Fig. 3). After defect nucleation, the transformation stagnates and two-phases coexist. In order to drive the transformation further, the hydrogen partial pressure must be increased. For the particle discussed previously, the complete transformation after the return to 0 mbar $pH_2$ occurred at 81 mbar $pH_2$ (Fig. 3a). For a second particle, after the return to 0 mbar $pH_2$, 97 mbar $pH_2$ was required to fully transform the particle (Fig. 3b). In both cases, the defect network acts as a barrier to completing the phase transformation. This barrier can be overcome by a sufficiently high hydrogen partial pressure.

## Discussion

Nanoparticles show dramatically different properties relative to their bulk counterparts, and these properties are strongly affected by the presence of crystallographic defects. Here we have shown that Pd nanoparticles possess the unique ability to self-heal these crystallographic defects to varying degrees, which may help to explain the improved properties of nanostructured systems, including advanced battery cathode, anode, and hydrogen storage systems. Future work could address, in more detail, the mechanistic picture of dislocation healing by utilizing the experiments and analysis tools developed in this work. Furthermore, more particles should be measured to understand their heterogeneous response. Our results resolve the formation and healing of structural defects during structural phase transformations at the single nanoparticle level and offer an additional perspective as to how nanoparticles can exhibit different properties from their bulk counterparts.

## Methods

**Pd Nanoparticle Synthesis**. Pd nanoparticles were formed via a dewetting procedure. 25 nm of Pd was deposited by e-beam evaporation onto polished glassy carbon disks. The samples were annealed in a tube furnace at 400 °C for 2 h and then at 900 °C for 1 h under a $N_2$ atmosphere.

**Bragg Coherent Diffractive Imaging experiment details**. Experiments were performed at Sector 34-ID-C of the Advanced Photon Source at Argonne National Laboratory. A double crystal monochromator was used to select $E = 8.919$ keV X-rays with 1 eV bandwidth and longitudinal coherence length of 0.7 μm. A set of Kirkpatrick-Baez mirrors was used to focus the beam to $0.5 \times 0.6$ μm$^2$ (vertical x horizontal). The rocking curve around the (111) Pd Bragg peak was collected by recording 2D coherent diffraction patterns with an X-ray sensitive area detector (Medipix2/Timepix, $512 \times 512$ pixels, each pixel 55 μm × 55 μm) around $2\theta = 33°$ ($\Delta\theta = \pm 0.2°$). The camera does automatic background subtraction. It was placed a distance of 0.5 m away from the sample and an evacuated flight tube was inserted between the sample and the camera. A total of 61 patterns were collected for a single 3D rocking scan. The patterns were collected continuously while the Pd nanocrystals transformed. Each 3D scan takes between 5–6 min. To transform the crystals, a 4% mole fraction $H_2$ gas in He was flowed at varying rates using a 1000 SCCM mass flow controller. A separate flow controller was used to dilute the $H_2$ gas with $N_2$ gas in order to reach the lower hydrogen partial pressure steps. The total pressure inside the cell was measured using a capacitance manometer.

**Phase retrieval**. The phase retrieval code is adapted from published work. The difference map and error reduction algorithms were used for all reconstructions. 2200 total iterations, consisting of alternating 90 iterations of the relaxed difference map algorithm with 10 iterations of the error reduction algorithm, were run for 5 reconstructions beginning from random phases. The best reconstruction, quantified by the smallest sharpness metric, was then used in conjunction with another random phase start as a seed for another 10 random starts. The sharpness metric is the sum of the absolute value of the reconstruction raised to the 4th power. 5 generations were used in this guided algorithm. The final resolution of 20–30 nm was computed from the phase retrieval transfer function, an example of which is shown in Supplementary Fig. 7.

**Data availability**. The data reported in this paper are available upon request. All code, including the reconstruction algorithm, is also available upon request.

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

## Acknowledgements

Design of the hydriding phase transformation experiment and image analysis was supported by the DOE Office of Science, Office of Basic Energy Sciences, Division of Materials Sciences and Engineering. Sample synthesis was supported by the National Science Foundation CHE-1565945. This research used resources of the Advanced Photon Source, a U.S. Department of Energy (DOE) Office of Science User Facility operated for the DOE Office of Science by Argonne National Laboratory under Contract No. DE-AC02-06CH11357. We thank Ross Harder and Wonsuk Cha for maintaining the beamline. We thank Evan Maxey for his help in the cell design. Finally, we thank the staff at the Advanced Photon Source for their support.

## Author contributions

A.U.: Designed the experiment. A.Y.: Made the samples. A.U. and A.Y.: Performed the experiments. A.U.: Wrote the manuscript. All authors contributed to editing the manuscript.

## Additional information

**Competing interests:** The authors declare no competing financial interests.

