## [Peer Review File · Nature Communications]

Reviewers' comments:

Reviewer #1 (Remarks to the Author):

This paper reports Bragg coherent diffraction measurements of Pd nano-particles during the hydriding phase transformation. As the phase transformation progresses dislocations appear. The main result of the paper is that these defects disappear again when the phase transformation is reversed. For the particles under study isotherms show multiple plateaus and interpreted in terms of dislocation networks acting as obstacles to phase transformation.

The observation that dislocations formed during the hydriding phase transition are removed upon subsequent reversal of the phase transition is interesting. Unfortunately no attempt is made to explore the underlying mechanisms governing this behaviour. Yet this mechanistic insight is essential for the optimisation of nano-particle performance.

A number of important questions arise from the data:

1. What is the mechanism of dislocation nucleation? In Fig. 1 several small loops in the centre of the hydrogen poor phase are shown. Why are small shear loops seen rather than extended dislocation lines that propagate through the crystal? Where do these dislocations nucleate? Is it at the phase interface as expected?
2. One of the loops in Fig. 1 (b), second image from the left, appears to not only contain a shear portion but also a prismatic part. This is very surprising. What could be the origin of the prismatic part? How can it be ascertained that this is not an artefact of the phase retrieval?
3. What is the mechanism by which dislocations are removed from the crystal after de-hydriding? Is it the collapse of dislocation loops? Or do dislocations escape to the crystal surface? What is the distinguishing feature that leads to some loops being retained, whilst others are lost? Which aspect of crystal size is most important for this defect removal?

Since the authors have coherent diffraction data for the full time history of this sample (in the supplementary material they state that scans were collected continuously with about a 6 minute interval), they should be able to address all these questions.

Several minor points must also be addressed:

1. Line 85 – the statement that the crystal must be defect free since no phase vortices are seen is incorrect. Dislocations will only show up when $q \cdot b$ is non zero. Furthermore small point defects, e.g. vacancy and interstitial clusters below the measurement resolution of the measurement, will not give rise to phase vortices, but rather more subtle phase features.
2. The sentence in lines 73, 74 is repeated in line 99.
3. In the sentence in line 130, 131 it is not clear what is meant by full and partial loops. Furthermore the relevance of this sentence to the rest of the discussion is unclear.
4. The approach used to calculate isotherms is not clear. A more detailed description must be provided.
5. In figure 3 it would help to include points recorded at 0 Torr during the dehydriding part of the

cycle.

6. It is not clear how the information provided in lines 155 to 157 is relevant to the discussion.

Reviewer #2 (Remarks to the Author):

Referee comment

Manuscript by A. Ulvestad and A. Yau: "The self-healing of defects in nanoparticles: insights from the hydriding phase transformation"

In this manuscript the authors performed Bragg coherent diffraction imaging (BCDI) studies of Pd nanoparticles in hydrogen environment at different hydrogen pressure. It was observed creation of defects in the nanoparticles of about 400 nm in size and then subsequent healing at zero pressure.

I have few points that should be commented by the authors:

1. Similar experiment was reported recently by the same first author (Ref. 14 of the manuscript). In that manuscript it was made a strong statement that dislocations formed in the nanoparticle are occulting in the boundary between the high hydrogen region and low hydrogen region. However from the present results we see that this is not the case and dislocation network is going through the particle at high hydrogen pressure (see Fig. 2 of the manuscript).

2. I think statistics of observation is not sufficient to make any claims on behavior of particles. As we can see from the supplementary materials already 2-nd particle does not reduce the strain at zero pressure (See Suppl. Fig. 4d).

3. I do not understand how Fig. 3 was produced and how measurements were performed at different pressures. For example in Fig. 3a at pressure of 60 Torr, it is shown continuous variation of strain at the same pressure, how it was obtained?

4. One technical issue: scale bars should be changed to 100 nm scale, instead of 105 nm.

Finally, my impression is that added value to the knowledge of the process of hydrogen storage in nanoparticles is not sufficient for the publication of this manuscript in Nature Communication journal. My suggestion to the authors would be to go with the resubmission of the manuscript to a more specialized journal.

Reviewer #3 (Remarks to the Author):

This paper is a beautiful study of defect formation and annihilation in single nanostructured materials undergoing phase transitions in a reactive environment.

Although the paper is novel, clearly written and its conclusions are solid, I have a few comments and suggestions that I would like the authors to consider before recommending publication.

LINE 12

The sentence "However, this current understanding is incomplete." is very vague and should be qualified further.

LINE 27

References 1 and 2 seem rather randomly chosen.

LINE 33

What do the authors exactly mean with “nanostructuring improves their performance”?

LINE 35

The sentence “these explanations miss the amazing ability of nanoparticles to self-heal crystallographic defects” seems rather exaggerated. The present work shows beautiful and compelling proof of defect healing in nanostructured materials, but some evidence of improved crystallinity upon cycling of nanocrystals has already been presented, for example by Narayan et al., Nature Communications 8, 14020 (2017), where they write “Cycling experiments consisting of two loading and unloading processes confirm that the particle improves its crystal quality during the α to β phase transformation. The ability of a nanoparticle to remove prior imperfections allows it to be an effective medium for energy storage, as it is able to maintain a high degree of crystallinity during the cycling process. Our results demonstrate the utility of nanoscaling for solute-based phase transformations—the small size allows for defects to be pushed out of the particle, which is not possible in the bulk”.

LINE 44

Reference 13 is an ensemble study and it does not address “individual Pd nanoparticles”.

LINE 46

On the sentence “As such, they never observed the amazing phenomenon of defect healing” see my comment on LINE 35.

LINE 60

The caption of Figure 1 should explicitly mention that what is plotted are displacement field maps. The choice of a scale bar of “105 nm”, which is repeated in other figures, seems rather unusual.

LINE 65

I don't see the black plane in the figure.

LINE 67

Is there a specific reason to choose “torr” as the unit of pressure? I would personally recommend the use of more common units in the field, such as Pa, mbar, or bar.

LINE 93

I find the following observation rather remarkable: “These are due to dislocations, and it is clear that these dislocations exist in the particle center that remains in the hydrogen-poor phase”. The nucleation of dislocations far away from the alpha/beta interface (see for example z3 at 45.7 torr) beautifully shows the long-range character of elastic strain in the Pd/H₂ system.

LINE 99

The authors write “More shockingly, the dislocations that were induced in the particle's center have healed”. Is it at all possible that the dislocations would have healed already upon full hydrogenation, had the authors increased the pressure even further? This would be consistent with the observation of Narayan et al., Nature Communications 8, 14020 (2017), where the authors find an increased crystallinity upon hydrogenation (their Figure 4b). Since the authors here only map the alpha phase, it seems to me impossible to rule out this possibility and it should perhaps be mentioned.

LINE 107

The authors write "The generation and subsequent healing of the defects can also be seen directly in the coherent diffraction data (Supplementary Fig. 3)". I think this needs more explanation: how is Figure 3 in the SI showing defect healing?

LINE 138

When the authors write "To construct these isotherms from the diffraction data, we consider two different regions", it would be helpful to specify that these two "regions" are before and after the alpha to beta phase transition.

LINE 139

The authors write "Before the phase transformation, the Pd lattice expands linearly with the hydrogen concentration". Technically, this is true independently of the hydrogen concentration, not just "before the phase transformation" (see for example Figure 3.10 in H. Peisl, "Lattice strains due to hydrogen in Metals", in Topics in Applied Physics, Hydrogen in Metals I, (ed. G. Alefeld and J. Völkl), Springer-Verlag (1978)).

LINE 144

"the diffraction intensity is proportional to the square of the number of atoms in the hydrogen-poor phase". This sentence may need more justification, or at least a reference.

LINE 163

The authors write "For the particle discussed previously, the complete transformation occurred at 61 Torr p_{H2} (Fig. 3a)". This sentence seems to be in contradiction with the previous sentence "the region containing many dislocations remaining in the hydrogen-poor phase throughout". Did the centre of the particle fully hydrogenate, as the isotherm seems to suggest? If so, my comment on line 99 is even more relevant.

LINE 175

I find the final sentence "Our results fundamentally change our understanding of nanoscale plasticity and offer a new perspective as to how and why nanoparticles are different from their bulk counterparts" unnecessary. The present work beautifully shows the formation and healing of structural defects in single nanostructures undergoing complex phase transformations: to claim that this "fundamentally change our understanding of nanoscale plasticity" is too strong.

Reviewer concerns are in black text, our responses are in blue, manuscript changes indicated in green.

Reviewer #1

1. What is the mechanism of dislocation nucleation?

This is an interesting question that was not sufficiently addressed. We have added the following to the text:

Dislocation nucleation is driven by strain energy relaxation. The lattice mismatch between the hydrogen-rich and the hydrogen-poor phases is approximately 3.5%. As such, it becomes energetically favorable above a critical size to relieve the strain energy through dislocation nucleation. For the idealized spherical particle with a spherical shell of hydrogen-rich phase, misfit dislocation loops are introduced at the phase boundary. However, this description is a rough approximation of the experimental observations. The two-phase morphology is more similar to a spherical cap with a planar interface between the two phases. Although the phase morphology is different, strain energy relief still drives dislocation nucleation.

In Fig. 1 several small loops in the centre of the hydrogen poor phase are shown. Why are small shear loops seen rather than extended dislocation lines that propagate through the crystal?

We have added the following to the manuscript:

The dislocation lines form an extended network throughout the crystal.

Where do these dislocations nucleate? Is it at the phase interface as expected?

We have added the following to the manuscript:

In this work, we cannot comment on the dislocation nucleation mechanism^{18–20} as it occurs too rapidly relative to the experimental measurement time. However, we can observe the stability and mobility of dislocations after they are nucleated. As will be shown later using the 3D view, the dislocation lines form an extended network throughout the crystal that connects to the phase boundary.

2. One of the loops in Fig. 1 (b), second image from the left, appears to not only contain a shear portion but also a prismatic part. This is very surprising. What could be the origin of the prismatic part? How can it be ascertained that this is not an artefact of the phase retrieval?

We do not believe there is a prismatic part. Because ascertaining the geometry from 2D images is challenging, we have uploaded movies that show the images in Fig. 2 rotating in 3D.

3. What is the mechanism by which dislocations are removed from the crystal after de-hydrogenating? Is it the collapse of dislocation loops? Or do dislocations escape to the crystal

surface? What is the distinguishing feature that leads to some loops being retained, whilst others are lost? Which aspect of crystal size is most important for this defect removal?

We can speculate based on the observed data that dislocation loops are able to collapse during the dehydriding process. We see in Fig. 1d that the only loop remaining after the reverse transformation is near the top boundary of the particle. This could imply that the surface has a stabilizing effect on the dislocation loops. We have added the following to the main text:

Based on this data, we speculate that the dislocation loops in the particle interior are able to collapse during the dehydriding phase transformation. The remaining loop that is near the surface could be stabilized by a dislocation loop-surface interaction.

Since the authors have coherent diffraction data for the full time history of this sample (in the supplementary material they state that scans were collected continuously with about a 6 minute interval), they should be able to address all these questions.

Unfortunately we do not have data during the dehydriding at 0 mbar. Here is a snapshot of the timestamps showing the gap:

```
drwxrwsr-x  2 epix34id xsdmc  4096 Mar 19 06:24 Ulvestad317b_S1723/
drwxrwsr-x  2 epix34id xsdmc  4096 Mar 19 09:48 Ulvestad317b_S1727/
```

Several minor points must also be addressed:

1. Line 85 – the statement that the crystal must be defect free since no phase vortices are seen is incorrect. Dislocations will only show up when $q \cdot b$ is non zero. Furthermore small point defects, e.g. vacancy and interstitial clusters below the measurement resolution of the measurement, will not give rise to phase vortices, but rather more subtle phase features.

This is an excellent point. While it is true that $q \cdot b \approx 0$ is the condition for observing a screw dislocation, the more general condition is that dislocations are invisible if $q \cdot u = 0$ where u is the displacement field generated by the dislocation. This distinction needs to be made as edge dislocations generate displacements perpendicular to their Burger's vector. We have revised the manuscript as follows:

Most importantly, the displacement field is free of any spiral regions in which the displacement field varies from $-d_{111}/2$ to $+d_{111}/2$ (displacement field vortices) and thus the particle has no dislocations that are visible at this scattering condition. For a dislocation to be visible at a particular scattering vector, there must be a non-zero projection of the displacement field onto the scattering vector. Additionally, defects such as vacancy and interstitial clusters are below the measurement resolution.

2. The sentence in lines 73, 74 is repeated in line 99.

We have reworded line 99.

3. In the sentence in line 130, 131 it is not clear what is meant by full and partial loops. Furthermore the relevance of this sentence to the rest of the discussion is unclear.

We have removed this sentence as it is not essential to the discussion.

4. The approach used to calculate isotherms is not clear. A more detailed description must be provided.

We have added the following:

In the solid solution regime before the phase transformation, the hydrogen-poor Pd lattice expands linearly with the hydrogen concentration and thus the percent strain is linearly related to the hydrogen concentration. Using the Bragg peak location to define the average lattice constant, the percent strain can then be calculated via

$$\epsilon = 100 * \frac{a_{p_{H_2}} - a_0}{a_0}$$

where a_0 is the initial average lattice constant before any hydrogen exposure and $a_{p_{H_2}}$ is the average lattice constant at a given pressure. In the case of **Figure 3a**, the solid solution regime lasts through 46.5 mbar p_{H_2} .

Once the hydrogen-rich phase begins to appear, the average lattice constant of the hydrogen-poor phase is no longer proportional to the hydrogen concentration. In this regime, the percent strain is calculated via

$$\epsilon = 100 * (x_\alpha \frac{3.896 - 3.89}{3.89} + (1 - x_\alpha) \frac{4.025 - 3.89}{3.89})$$

where x_α is the fraction remaining in the hydrogen-poor phase. This is determined using the square root of the ratio of the total scattering intensity at a given partial pressure to the total scattering intensity in the initial hydrogen-free state.

5. In figure 3 it would help to include points recorded at 0 Torr during the dehydrating part of the cycle.

We have produced a plot including the 0 mbar point and are including it as a Supplementary Figure. It is reproduced below for convenience.

a) Particle previously discussed

b) Particle 2

Supplementary Figure 6. Pressure-strain isotherms for the particles discussed in the main text. The return to 0 mbar is shown in this Figure.

6. It is not clear how the information provided in lines 155 to 157 is relevant to the discussion.

We have clarified the discussion as follows:

After reversing the phase transformation by applying 0 mbar for 4 hours, the hydrogen partial pressure is incremented to 81 mbar. The isotherm including the 0 mbar data point is shown in **Supplementary Fig. 6**.

Reviewer #2 (Remarks to the Author):

1. Similar experiment was reported recently by the same first author (Ref. 14 of the manuscript). In that manuscript it was made a strong statement that dislocations formed in the nanoparticle are occulting in the boundary between the high hydrogen region and low hydrogen region. However from the present results we see that this is not the case and dislocation network is going through the particle at high hydrogen pressure (see Fig. 2 of the manuscript).

This is an important point that we did not explain. In Ref. 14, the hydrogen partial pressure was incremented in one step from 0 to 25 Torr. 25 Torr is very close to the pressure required to drive the transformation for the particle size investigated in Ref. 14. This pressure was chosen so that the phase transformation proceeded very slowly and there was only enough driving force to nucleate a few dislocations. In this work, we incremented the partial pressure through many steps and substantially exceeded the upper spinodal pressure. With this additional driving force, it is expected that more dislocations will occur, consistent with the results in this work. We have added the following to the manuscript to make this distinction clear:

The primary difference between this work and that of ref. ¹⁴ is the amount of driving force supplied by the hydrogen partial pressure. In ref. ¹⁴, the hydrogen partial pressure was very close to the upper spinodal pressure and consequently the driving force for the phase transformation is small. In this work, the partial pressure is incremented well beyond the upper spinodal pressure and consequently the driving force is large.

2. I think statistics of observation is not sufficient to make any claims on behavior of particles. As we can see from the supplementary materials already 2nd particle does not reduce the strain at zero pressure (See Suppl. Fig. 4d).

The first particle (discussed in the main text) exhibits strain and dislocations after 4 hours at 0 mbar, consistent with the 2nd particle shown in the supplemental material. In particular, slice z_3 in both particles shows a few remaining dislocations after 4 hrs at 0 mbar. These two results are quite consistent with one another.

3. I do not understand how Fig. 3 was produced and how measurements were performed at different pressures. For example in Fig. 3a at pressure of 60 Torr, it is shown continuous variation of strain at the same pressure, how it was obtained?

This is a good point. A continuous variation of strain at the same pressure occurs during the phase transformation as more of the particle volume transforms from the hydrogen-poor phase to the hydrogen-rich phase. The section on the isotherm calculation has been updated and the relevant section is:

Once the hydrogen-rich phase begins to appear, the average lattice constant of the hydrogen-poor phase is no longer proportional to the hydrogen concentration. In this regime, the percent strain is calculated via

$$\epsilon = 100 * (x_{\alpha} \frac{3.896 - 3.89}{3.89} + (1 - x_{\alpha}) \frac{4.025 - 3.89}{3.89})$$

where x_{α} is the fraction remaining in the hydrogen-poor phase. This is determined using the square root of the ratio of the total scattering intensity at a given partial pressure to the total scattering intensity in the initial hydrogen-free state.

4. One technical issue: scale bars should be changed to 100 nm scale, instead of 105 nm.

We have changed all the scalebars, thank you.

Finally, my impression is that added value to the knowledge of the process of hydrogen storage in nanoparticles is not sufficient for the publication of this manuscript in Nature Communication journal.

The primary novel result of this work is that dislocations in large nanoparticles that are induced by the hydriding phase transformation can be subsequently healed by the dehydriding phase transformation. This is quite new and exciting, and has never been observed. It is not expected from the traditional picture of irreversible dislocation nucleation. These results add a new dimension to our understanding of why nanoparticles often show improved characteristics such as better cyclability. Furthermore, the results are likely applicable to other systems that exhibit solute-induced phase transformations, such as intercalating battery cathodes.

Reviewer #3 (Remarks to the Author):

LINE 12

The sentence “However, this current understanding is incomplete.” is very vague and should be qualified further.

Our point can be made without this sentence so we have removed it.

LINE 27

References 1 and 2 seem rather randomly chosen.

We have added more references to works that explore the effect of nanosizing on various properties.

LINE 33

What do the authors exactly mean with “nanostructuring improves their performance”?

We have changed the text to the following:

At the same time, it is known that nanostructuring can improve rate capabilities, lifetime, and reduce overpotential^{3,9}.

LINE 35

The sentence “these explanations miss the amazing ability of nanoparticles to self-heal crystallographic defects” seems rather exaggerated. The present work shows beautiful and compelling proof of defect healing in nanostructured materials, but some evidence of improved crystallinity upon cycling of nanocrystals has already been presented, for example by Narayan et al., Nature Communications 8, 14020 (2017), where they write “Cycling experiments consisting of two loading and unloading processes confirm that the particle improves its crystal quality during the α to β phase transformation. The ability of a nanoparticle to remove prior imperfections allows it to be an effective medium for energy storage, as it is able to maintain a high degree of crystallinity during the cycling process. Our results demonstrate the utility of nanoscaling for solute-based phase transformations—the small size allows for defects to be pushed out of the particle, which is not possible in the

bulk”.

This is an excellent point. We have changed the manuscript as follows:

Recent research has shown that the crystal quality of Pd nanoparticles can improve as a result of the phase transformation¹⁷.

LINE 44

Reference 13 is an ensemble study and it does not address “individual Pd nanoparticles”.

We have updated the manuscript as follows:

Many previous experimental studies have used electron microscopy^{12,16,17} and plasmonic techniques¹⁸ to track individual Pd nanoparticles during the hydriding phase transformation. Ensemble studies have been performed with luminescence-based methods¹⁹.

LINE 46

On the sentence “As such, they never observed the amazing phenomenon of defect healing” see my comment on LINE 35.

We have removed this statement.

LINE 60

The caption of Figure 1 should explicitly mention that what is plotted are displacement field maps. The choice of a scale bar of “105 nm”, which is repeated in other figures, seems rather unusual.

Thank you for pointing out the error. We have corrected the caption. We have also updated all scalebars to be 100 nm.

LINE 65

I don't see the black plane in the figure.

We have updated the manuscript as follows:

The grey plane in the 3D view indicates the substrate.

LINE 67

Is there a specific reason to choose “torr” as the unit of pressure? I would personally recommend the use of more common units in the field, such as Pa, mbar, or bar.

We have updated the units to mbar.

LINE 99

The authors write “More shockingly, the dislocations that were induced in the particle’s center have healed”. Is it at all possible that the dislocations would have healed already upon full hydrogenation, had the authors increased the pressure even further? This would be consistent with the observation of Narayan et al., Nature Communications 8, 14020 (2017), where the authors find an increased crystallinity upon hydrogenation (their Figure 4b). Since the authors here only map the alpha phase, it seems to me impossible to rule out this possibility and it should perhaps be mentioned.

We weren’t clear on this point. We updated the text and added a supplemental figure, which is reproduced below for convenience.

Finally, we drove the phase transformation to completion by again applying a p_{H_2} of 81 mbar. Reconstructions of the coherent diffraction data (**Supplementary Fig. 3**) are not reproducible due to the complexity of the data, which implies the particle has a high density of defects such as dislocations. If the particle were able to heal defects during the complete transformation, we would expect a coherent diffraction pattern very similar to the pattern of the initial state but this is not the case.

Supplementary Figure 3. Coherent diffraction data for Pd particle discussed in the main text. **a**, The as-synthesized state. **b**, After the complete transformation. In **b** there is a lack of centrosymmetry, a clearly defined central maximum, and the spread in the diffraction pattern is much larger than in **a**. All of these features are consistent with a highly defective particle.

LINE 107

The authors write “The generation and subsequent healing of the defects can also be seen directly in the coherent diffraction data (Supplementary Fig. 3)”. I think this needs more explanation: how is Figure 3 in the SI showing defect healing?

This is not obvious and so we have changed the manuscript text to the following:

Defect generation results in significant changes to the coherent diffraction data that then mostly reverse during the dehydrogenating transformation (**Supplementary Fig. 4**).

LINE 138

When the authors write “To construct these isotherms from the diffraction data, we consider two different regions”, it would be helpful to specify that these two “regions” are before and after the alpha to beta phase transition.

This is a good point and we have updated the isotherm calculation description.

LINE 139

The authors write “Before the phase transformation, the Pd lattice expands linearly with the hydrogen concentration”. Technically, this is true independently of the hydrogen concentration, not just “before the phase transformation” (see for example Figure 3.10 in H. Peisl, “Lattice strains due to hydrogen in Metals”, in Topics in Applied Physics, Hydrogen in Metals I, (ed. G. Alefeld and J. Völkl), Springer-Verlag (1978)).

Thank you for catching this mistake. We have updated the manuscript as follows:

In the solid solution regime before the phase transformation, the hydrogen-poor phase of the Pd lattice expands linearly with the hydrogen concentration and thus the percent strain is linearly related to the hydrogen concentration.

LINE 144

“the diffraction intensity is proportional to the square of the number of atoms in the hydrogen-poor phase”. This sentence may need more justification, or at least a reference.

We have added a reference to the Elements of Modern X-ray Physics.

LINE 163

The authors write “For the particle discussed previously, the complete transformation occurred at 61 Torr p_{H2} (Fig. 3a)”. This sentence seems to be in contradiction with the previous sentence “the region containing many dislocations remaining in the hydrogen-poor phase throughout”. Did the centre of the particle fully hydrogenate, as the isotherm seems to suggest? If so, my comment on line 99 is even more relevant.

We clarified the text as follows:

For the particle discussed previously, the complete transformation after the return to 0

mbar $p\text{H}_2$ occurred at 81 mbar $p\text{H}_2$ (**Fig. 3a**). For a second particle, after the return to 0 mbar $p\text{H}_2$, 97 mbar $p\text{H}_2$ was required to fully transform the particle (**Fig. 3b**).

and

The ability to resolve the entire defect network in 3D detail shows that the dislocation lines induced during the partial forward transformation extend throughout the particle and are subsequently self-healed during the reverse transformation.

LINE 175

I find the final sentence “Our results fundamentally change our understanding of nanoscale plasticity and offer a new perspective as to how and why nanoparticles are different from their bulk counterparts” unnecessary. The present work beautifully shows the formation and healing of structural defects in single nanostructures undergoing complex phase transformations: to claim that this “fundamentally change our understanding of nanoscale plasticity” is too strong.

We have revised the manuscript as follows:

Our results resolve the formation and healing of structural defects during structural phase transformations at the single nanoparticle level and offer an additional perspective as to how and why nanoparticles are different from their bulk counterparts.

REVIEWERS' COMMENTS:

Reviewer #1 (Remarks to the Author):

The authors have addressed most of the referees' comments in sufficient detail. It is unfortunate that the crucial information, diffraction data during the dehydriding at 0 mbar, is missing. As the authors state, this makes it difficult to identify the mechanism of dislocation removal during the de-hydriding phase transition; the main result of this paper. This lack of mechanistic insight substantially lessens the impact of the reported results. As such I am not certain that this paper is suitable for Nature Communications, but rather a more specialized journal might be a better fit.

Reviewer #2 (Remarks to the Author):

The authors have clarified most of my questions.

Clearly, one of the most important one how general are results obtained in this work, or in other words, is there any statistical evidence for such generality, can not be answered in the frame of this work. Claims made on the result of evaluation of 2 particles are not sufficient. Especially the 2-nd particle does not completely relax as it is clearly seen in the presented results.

As such, I would suggest to authors to remove all claims on universality of the obtained results from the Abstract (Introduction) and Summary of the manuscript.

Reviewer #3 (Remarks to the Author):

All my comments have been addressed and I am happy to recommend the manuscript for publication.

Reviewer #1 (Remarks to the Author)

The authors have addressed most of the referees' comments in sufficient detail. It is unfortunate that the crucial information, diffraction data during the dehydriding at 0 mbar, is missing.

We have added the following to the text:

Unfortunately, the annihilation mechanism is unclear as measurements were not made continuously during the four hours at 0 mbar. However, future experiments will be able to address this issue using the tools and analysis developed in this work.

and

Future experiments can address the mechanism using the results of this work.

Reviewer #2 (Remarks to the Author):

The authors have clarified most of my questions.

Clearly, one of the most important one how general are results obtained in this work, or in other words, is there any statistical evidence for such generality, can not be answered in the frame of this work. Claims made on the result of evaluation of 2 particles are not sufficient. Especially the 2-nd particle does not completely relax as it is clearly seen in the presented results.

We added the following to the discussion:

While this particle does not relax as fully within the four hours at 0 mbar as the particle discussed in Fig. 1, that is probably due to slight differences in morphology. Consequently, there is likely variation in the degree to which particles can remove defects.

As such, I would suggest to authors to remove all claims on universality of the obtained results from the Abstract (Introduction) and Summary of the manuscript.

In the abstract we removed claims on universality of obtained results by changing the sentence "Here we show that nanoparticles also have the ability to self-heal defects in their crystal structures, which contributes to the improved properties of nanostructured systems such as batteries and metal hydrides." to the following sentence:

Here we show that Pd nanoparticles also have the ability to self-heal defects in their crystal structures.

and

Furthermore, more particles should be measured to understand their heterogeneous response.

We have also added the following to discuss what future work would be required:

However, future experiments will be able to address this issue using the tools and analysis developed in this work.